# Multi-Sensor Information Ensemble-Based Automatic Parking System for Vehicle Parallel/Nonparallel Initial State

**DOI:** 10.3390/s21072261

**Published:** 2021-03-24

**Authors:** Changhao Piao, Jun Zhang, KyungHi Chang, Yan Li, Mingjie Liu

**Affiliations:** 1School of Automation, Chongqing University of Posts and Telecommunications, Chongqing 400065, China; piaoch@cqupt.edu.cn (C.P.); s180302001@stu.cqupt.edu.cn (J.Z.); 2Department of Electrical and Computer Engineering, Inha University, 100 Inha-ro Michuhol-gu, Incheon 22212, Korea; khchang@inha.ac.kr (K.C.); leeyeon@inha.ac.kr (Y.L.)

**Keywords:** automatic parking system, parking space recognition, parking space matching, trajectory generation

## Abstract

The goal of automatic parking system is to accomplish the vehicle parking to the specified space automatically. It mainly includes parking space recognition, parking space matching, and trajectory generation. It has been developed enormously, but it is still a challenging work due to parking space recognition error and trajectory generation for vehicle nonparallel initial state with parking space. In this study, the authors propose multi-sensor information ensemble for parking space recognition and adaptive trajectory generation method, which is also robust to vehicle nonparallel initial state. Both simulation and real vehicle experiments are conducted to prove that the proposed method can improve the automatic parking system performance.

## 1. Introduction

Automatic parking systems (APS), as one of the main functions in advanced driver assistance systems (ADAS) and autonomous driving, is the key technology of transportation and traffic management [1,2,3]. It refers to the Internet of Things (IoT) [4] and artificial intelligence [5]. In addition, with the development of increased computing power, multi-sensor fusion technology has made a great contribution to smart parking systems.

To design an automatic parking system, it should consist of parking space recognition, parking space matching, trajectory generation [6,7,8], and vehicle control [9]. Parking space recognition [10,11] is one of the crucial components for APS, which is commonly realized based on machine vision technique. Following parking space recognition, parking space matching [12,13] requires the positioning information of the target parking space to guarantee the precision of the trajectory generation. From the trajectory generation perspective [14,15], it needs to establish the vehicle dynamics and obstacle avoidance models to generate available parking trajectory.

## 2. Related Works

This Section introduces related works that motivate this study: parking space recognition, parking space matching, trajectory generation, and vehicle control.

Jung et al. [16] describe a monocular vision-based parking-slot-markings recognition selection of automatic parking assist system. Hsu et al. [17] propose a configuration of the APS including sensors information fusion, position estimation, path planning, and tracking algorithm, which shows a good maneuver performance for vehicle. Ma et al. [18] develop an automatic parking system based on parking scene recognition. It introduces machine vision and pattern recognition techniques to intelligently recognize a vertical parking scenario, plan a reasonable parking path, develop a path tracking control strategy to improve the vehicle control automation, and explore a highly intelligent automatic parking technology road map. Liu et al. [19] focus on positioning accuracy and stability research, which is crucial for automatic parking system. They propose a new adaptive robust four-wheel calculation positioning method to address the problems of low precision and poor stability. It is based on the local outlier factor detection algorithm, in which adaptive diagnosis and compensation are made for data anomalies of four-wheel speedometers. The preview correction method is used for parking path planning in [20]. This method can not only detect the curvature outliers in the parking path, but also correct and optimize a reasonable parking trajectory in advance. Based on the Dijkstra algorithm, a scheme that can consider dynamic influence factors [21] is proposed to solve the lane occupancy caused by parking. Zhao et al. [22] designed an indoor automatic parking system based on indoor positioning and navigation technology. The power-aware path planning algorithm was proposed in [23]. This method determines the best parking place in the automatic parking system, calculates the best path, and can greatly reduce power consumption. Considering the possibility of collision between the car body and obstacles in the parking space and the continuity of parking needs, a Bezier curve is used in [24] to fit the trajectory of automatic parking. An improved genetic algorithm and time-enhanced A* algorithm trajectory calculation method [25] is used to solve the problem of high-density parking lot path planning. This method has been effectively improved in terms of driving distance and safety. In [26], parallel line pairs are extracted from the AVM image to detect the dividing line. According to the geometric constraints of the parking spaces, the separation lines are paired to generate candidate parking spaces. By using line and corner features to identify where they entered, and using ultrasonic sensors to classify their positions, the candidates are determined. In [27], a novel detection method based on deep convolutional neural network is proposed, and the largest data set in the field of parking space recognition is established to overcome various unpredictable factors affecting parking spaces. The parking space detection method based on the direction entry line regression and classification of the deep convolutional neural network [28] can easily detect parking spaces of different shapes from different angles. The parking space marking detection method based on the geometric features of parking spaces [29] mainly includes separation line detection and parking space entry detection. This method can identify typical vertical and parallel rectangular parking spaces with high accuracy.

Most of the existing methods are applicable when the initial state of a vehicle is parallel to the target space and can only generate a fixed-point trajectory [30]. Further, narrow parking area and limited parking points are also serious problems to APS. To overcome these limitations, we design an automatic parking system design, which can realize precision parking with different vehicle initial states (parallel/nonparallel to the target parking space). The primary study objective is to develop a novel APS in the engineering practice context as it overcomes the aforementioned shortcomings. This paper adopts the idea of data fusion, and then designs a multi-sensor data fusion method. The parking space matching uses the visual positioning method, and this paper also uses the experimental method to correct the image distortion caused by the optical lens. The trajectory calculation method is a traditional geometric method because it has an irreplaceable and efficient calculation speed. To sum up, the main contribution of this paper can be summarized as follows: (I) Multi-sensor information is fused to recognize the parking space, which can not only enhance the recognition performance, but also provide support for parking space matching. (II) A self-adaptive trajectory generation method is proposed to satisfy both parallel and nonparallel initial state. Moreover, some simulation and real vehicle experiments are conducted to demonstrate our method. Through the scheme designed in this paper, the probability of correct parking space matching reaches 94%, the maximum matching error of parking space is only 5 cm, and the parking success rate is as high as 90%.

The remainder of this paper is organized into four sections. Section 2 introduces related works. The proposed method is described in detail in Section 3. Section 4 designs the simulation and real vehicle experiment. Section 5 demonstrates the experimental results and contains a discussion on the specific analysis. Section 6 provides concluding remarks.

## 3. Automatic Parking System

### 3.1. Multi-Sensor Information Ensemble-Based Parking Space Recognition

Parking space recognition, as the first step for automatic parking system, highly depends on the information collected by sensors such as ultrasonic or cameras. Multi-sensor information ensemble can improve recognition performance. The parking space consists of an upper edge and a lower one. Ultrasonic sensors, which are installed on the right-hand side of the vehicle, are used for edge detection. The installed ultrasonic model is LGCB1000-18GM-D1/D2-V15, and the detection range is 70−1000 mm. The mimic diagram of a parallel parking spot is shown in Figure 1.

A range including target parking space is obtained by ultrasonic sensors, and the image collected by sensors is smoothed by filtering. Thereafter, the distance between the upper and lower edges is measured, and also the displacement of the vehicle to the target parking spot. Subsequently, the data measured by the two sensors are fused through the data fusion algorithm proposed in this paper.

Parking spot detection is of great importance to parking space recognition. We design a parking spot detection algorithm consisting of ultrasonic range determination, edge detection, and spot length calculation, as shown in Figure 2. In this study, polling-driven mechanism is introduced to trigger the sensors, which means one of the sensors is firstly triggered to produce ultrasonic waves with a fixed cycle. If the echoes are received by the triggered one, then the system polls the next channel for the other sensor.

The process is repeated in cycles. Time of flight (ToF) is calculated by the time capture register of the micro control unit (MCU), and the measured distance is obtained as follows:(1)$$d=\frac{v_{s}\cdot t}{2}$$
where $v_{s}=340\;m/s$ at 20 °C; t is the travel time.

Edge detection, which is crucial for the accuracy of parking spot detection, comprises upper edge detection and lower edge detection. In the case of parallel parking, the threshold of distance hop, which includes the hop threshold of the upper $H_{upper}$ and lower $H_{lower}$ edges, is calculated based on the threshold of the ultrasonic sensors $D_{th}$ and the cross range $D_{cross}$ between the target vehicle and parked vehicles, which is adjacent to the target parking spot. The threshold is determined by the width of the vehicle $W_{vehicle}$ and maximum range of the ultrasonic sensor $D_{\max}$.
(2)$$D_{th}=W_{vehicle}+D_{\max}$$

The edge thresholds are expressed as follows:(3)$$\{\begin{array}{c} H_{upper}=D_{th}-D_{cross}\\ H_{lower}=D_{cross}-D_{th} \end{array}$$
where $D_{th}-W_{vehicle}-D_{cross}\leq H_{upper}<D_{th}-W_{vehicle}$ and $W_{vehicle}-D_{th}\leq H_{lower}<W_{vehicle}-D_{th}+D_{cross}$.

The spot length is calculated by the driving speed and time:(4)$$L=\overset{N}{\underset{k=1}{\sum }}V(k)*T$$
where *N* is the cumulative number of vehicle displacements; *V*(*k*) is the running speed in cycle *k* Further, and *T* is the cycle time.

In multi-sensor information ensemble, a similarity model fusion is proposed. Compared with other methods such as D-S theory [31], Bayes theory [32], Kalman filter [33], and optimal statistical decision [34], the proposed method is suitable for the situations where prior knowledge is not available. In our method, multiple sensors are used to measure the same object, and the measured values are independent of each other. The process of information ensemble can be divided into several phases, as shown in Figure 3. The ensemble data sources are from the referential spot length and the measured values by two sensors. In the process of sensor measurement, the measurement result will be inaccurate due to the influence of the environment. Therefore, Formula (5) is used to compensate the measurement results of the ultrasonic sensor. Referential spot length is calculated by the difference between the average error of the sensors and corrected error. It can improve the data ensemble accuracy.
(5)$$L_{ref}=\frac{L_{1}+L_{2}}{2}-E$$
where $L_{1}$ and $L_{2}$ are the measured lengths of sensors 1 and 2, respectively; E is the corrected error computed using multiple linear regression model.

To quantify the similarity among the sensors at a particular moment, exponential decay function (EDF) [18] is employed to calculate the similarity and construct the similarity matrix. The conventional EDF is expressed as below:(6)$$S_{ij}(k)=e^{-\lambda_{sys}{\left(z_{i}(k)-z_{j}(k)\right)}^{2}}$$
where $\lambda_{sys}$ is a hyper-parameter; $S_{ij}(k)$ is the similarity between observed values of the ith and jth sensors; and e is the natural base. The measured value of the ith sensor at time k is denoted as $z_{i}(k)$. If $z_{i}(k)$ and $z_{j}(k)$ are considerably different, then the similarity between the ith and jth sensors is weak. Due to real world application, parking spot detection emphasizes real-time implementation in the embedded system. To this end, we define a new EDF function to avoid setting the experience value $\lambda_{sys}$ manually, which can brief the calculation process.
(7)$$S_{ij}(k)={\left(1+\frac{1}{N}\right)}^{-N|z_{i}(k)-z_{j}(k)|}$$
where N is the number of fused data sources; in our study, it is 3, including the referential length, data of sensor 1, and data of sensor 2. The similarity matrix at moment k can be expressed as follows:(8)$$S(k)=\left[\begin{array}{cccc} 1 & s_{12}(k) & \cdots & s_{1n}(k)\\ s_{21}(k) & 1 & \cdots & s_{2n}(k)\\ \vdots & \vdots & \vdots & \vdots \\ s_{n1}(k) & s_{n2}(k) & \cdots & 1 \end{array}\right]$$
where n is the number of fused data sources. It combines the similarity among sensors as a matrix where each line indicates the support degree among the sensors. $\overset{n}{\underset{j=1}{\sum }}s_{ij}(k)$ indicates the consistency among the ith sensor and others. The higher the value is, the more consistent it is. Therefore, the consistency measurement approach is defined as the assessment criteria:(9)$$c_{i}(k)=\frac{\overset{n}{\underset{j=1}{\sum }}s_{ij}(k)}{n}$$
where $c_{i}(k)$ is the proximity degree between the ith sensor and all the sensors (including the ith sensor) at k moment. Further, $\overset{n}{\underset{j=1}{\sum }}s_{ij}(k)$ is the sum of matrix line in Equation (8) and $0<c_{i}(k)\leq 1$. Target estimation only focuses on the consistency measurement $c_{i}(k)$ at a particular observation time. The target fusion value at time $k[\overset{\wedge }{sl}(k)]$ is expressed as follows:(10)$$\overset{\sum }{sl}(k)=\frac{\overset{n}{\underset{i=1}{\sum }}c_{i}(k)*z_{i}(k)}{\overset{n}{\underset{i=1}{\sum }}c_{i}(k)}$$
where $\overset{n}{\underset{i=1}{\sum }}c_{i}(k)$ is the sum of proximity degree of all the sensors at time *k*.

### 3.2. Parking Space Matching

The proposed parking space matching method uses image information collected by a wide-range camera installed in the rear portion of the vehicle. The purposes of parking space matching are to confirm the target parking space and to determine the target parking position as well as the distance with a vehicle [35,36]. The viewing angle range of the camera used is 120 degrees, the resolution is 640 × 480, and the chip is MT9V136. In our study, instead of employing an expensive positioning system such as GPS or inertial navigation, a convenient and accurate image ranging method is proposed. The common monocular camera is to solve object location problem based on the geometrical imaging model of the camera. The imaging model of the camera is shown in Figure 4. $P^{’}(u,v)$ and P(x,y) are the coordinate of the object in the world and image coordinate system, respectively; $S_{x}$ and $S_{y}$ are the width and height of the image. From the camera imaging geometry model, we obtain
(11)$$\left\{ \begin{array}{l} y=h_{cam}*\tan((90-\alpha_{cam})+((1-\frac{v}{S_{x}})*(\alpha_{cam}-\beta_{cam})))\\ x=y*((1-\frac{u}{S_{x}})*\theta_{cam})\\ L=\sqrt{x^{2}+y^{2}} \end{array}\right.$$
where L is the distance between the target point and camera. h_cam_ is the distance between the camera and the horizontal ground. This method can be successfully employed when the distance range is from 0.5–3.0 m. However, it does not satisfy the parking requirements that a minimum positioning distance range should be between 0.5 m and 9.0 m. In this study, the image coordinates and mapping points in the world coordinate are measured and some relationships are derived by analyzing and processing the measured data. There is an inverse proportional function relationship between the y-axis in an image coordinate and its corresponding longitudinal distance on the y-axis in the world coordinate system. For the x-axis, it has a proportional function relationship. The conversion relationship between (x,y) in image coordinates and (U,V) in world coordinates is given by Equation (12)
(12)$$\left\{ \begin{array}{l} X=0.84*y+6\\ V=\frac{22,139}{y+18}+174\\ U=320-320/X*x \end{array}\right.$$
where X represents the maximum horizontal distance that corresponds to line y in the world coordinates. The size of the captured images is 640×480. The center axis coordinates of the image are (320,0). The location data of the camera is shown in Table 1.

In this manner, a virtual space that has the same size with a parking space and the function of location can be established in the image systems. In this system, the virtual and real spaces can be matched by translation and rotation. Figure 5 shows the matching results.

### 3.3. Self-Adaptive Trajectory Generation for Vehicle Nonparallel/Parallel Initial State

To develop an automatic parking system, the direction of motion and trajectory of the vehicle should be known [37]. To this end, a vehicle kinematics model is established. Since automatic parking system always works with a low speed, the model can eliminate the possibility of sliding and lateral movements. The proposed vehicle kinematics model is shown in Figure 6.

The kinematic equations of the vehicle are expressed as:(13)$$\left\{ \begin{array}{l} \overset{•}{h_{c}}=(v_{c}/l_{c})*\tan s_{c}\\ \overset{•}{x_{rc}}=v_{c}*\cos h_{c}\\ \overset{•}{y_{rc}}=v_{c}*\sin h_{c} \end{array}\right.$$
where $h_{c}$ is the angle between the horizontal and the vehicle axle; $s_{c}$ is the angle between the vehicle front wheel and the vehicle axle. Further, $x_{rc}$ and $y_{rc}$ are the abscissa and ordinate of the rear axle center, respectively; $v_{c}$ and $l_{c}$ are the moving speed and wheelbase of the vehicle, respectively.

The trajectory is constituted from numerous sections of equal tangent arcs by analyzing the kinematics model. Typically, the trajectory is generated by using minimum radius method, in which the trajectory consists of two arcs with vehicle minimum radii. However, this kind of method requires that the vehicle body must be parallel to the parking space in the initial state. To overcome this limitation, a new trajectory generation method is proposed, which can process both parallel and nonparallel initial states. As shown in Figure 7, the vehicle rear axle center coordinates represent the entire vehicle trajectory, and each coordinate position of the vehicle can be calculated from the vehicle geometry and the current steering angle θ. The initial position of the vehicle is $S_{0}$, and the target position is $S_{d}$. The parking trajectory comprises arcs $S_{0}S_{1}$, $S_{1}S_{2}$, $S_{2}S_{3}$, and $S_{3}S_{d}$. The generated radius is different from the minimum radius; thus, its radii is unequal.

Considering the actual parking condition, the initial state of a vehicle is always nonparallel to parking spaces. It is always right or left skewed as shown in Figure 7. In our study, we consider these two conditions independently.

In the right skewed condition, as shown in Figure 7a, the car firstly turns left with center point $O_{1}$, and its radius is $R_{\min}$. When the rear axle center point reaches point $S_{1}$, the car moves straight up to point $S_{2}$. The car then turns right with center point $O_{1}$, and the corresponding radius is $R_{1}$. It maintains this state until arriving at point $S_{3}$. Finally, the car turns left with center point $O_{2}$ until reaching point $S_{d}$, and its radius is $R_{\min}$. From the geometric relationship and parking process perspective, the relationship between circles $O_{1}$ and $O_{2}$ can be expressed as follows: (14)$$\left\{ \begin{array}{l} d_{x}=x_{0}=(R_{1}+R_{\min})*(\cos\beta -\cos\alpha )\\ d_{y}=y_{0}=(R_{1}+R_{\min})*(\sin\beta +\sin\alpha ) \end{array}\right.$$
where $d_{x}$ and $d_{y}$ are the horizontal and vertical distances of the parking space obtained from the ultrasonic sensors, respectively; β is the initial attitude angle of the vehicle.

Then, α and $R_{1}$ can be calculated as follows:(15)$$\alpha =\sin^{-1}(\frac{y_{0}-R_{1}*\sin\beta }{R_{1}+R_{\min}})$$
(16)$$R_{1}=\frac{x_{0}^{2}+y_{0}^{2}-2R_{\min}*x_{0}}{2[\cos\beta *(R_{\min}-x_{0})-y_{0}*\sin\beta +R_{\min}]}$$

The trajectory for the right skewed initial state can be generated as follows:(17)$$\left\{ \begin{array}{l} {\left(x-R_{\min}*\beta \right)}^{2}+{\left(y-R_{\min}*\sin\beta \right)}^{2}=R_{\min}^{2}(0\leq y\leq y1)\\ x=R_{\min}*\cos\beta -R_{\min}(y1\leq y\leq y2)\\ {\left(x-R_{\min}*\cos\beta \right)}^{2}+[y-d_{y}-\sqrt{4R_{\min}*d_{x}-d_{x}^{2}}]=R_{\min}^{2}(y2\leq y\leq y3)\\ \left|x-(d_{x}-R_{\min}){|}^{2}+{\left(y-d_{y}\right)}^{2}=R_{\min}^{2}(y3\leq y\leq y_{d}\right) \end{array}\right.$$

The left skewed condition, as shown in Figure 7b, is similar with the right skewed condition. The relationship between circles $O_{1}$ and $O_{2}$ for left skewed condition can be expressed as follows:(18)$$\left\{ \begin{array}{l} d_{x}=(R_{1}+R_{\min})*(2-\cos\alpha -\cos\beta )\\ d_{y}=(R_{1}+R_{\min})*\sin\alpha +R_{1}*\sin\beta \end{array}\right.$$

The trajectory for the left skewed initial state can be generated as follows:(19)$$\left\{ \begin{array}{l} {\left(x-R_{\min}*\cos\beta \right)}^{2}+{\left(y-R_{\min}*\sin\beta \right)}^{2}=R_{\min}^{2}(0\leq y\leq y1)\\ x=R_{\min}-R_{\min}*\cos\beta (y1\leq y\leq y2)\\ {\left(x-2R_{\min}+R_{\min}*\cos\beta \right)}^{2}+[y-d_{y}-\sqrt{4R_{\min}*d_{x}-d_{x}^{2}}]=R_{\min}^{2}(y2\leq y\leq y3)\\ \left|x-(d_{x}-R_{\min}){|}^{2}+{\left(y-d_{y}\right)}^{2}=R_{\min}^{2}(y3\leq y\leq y_{d}\right) \end{array}\right.$$

### 3.4. The Workflow of the Automatic Parking System

The previous sections introduced the various components of the automatic parking system, but the entire system requires their cooperation to complete. The entire system first needs ultrasonic sensors and cameras to obtain parking space coordinate information, parking space size information, and obstacle location information, and then lock the matched parking space and generate a suitable parking track. In the parking process, the vehicle control system generates angle control signals and vehicle speed control signals based on the fuzzy control algorithm [38,39], transmits them to the electric power assist system and the vehicle speed control system, and ends the parking process when it reaches the end of the target trajectory. In order for the driver to operate more conveniently, the driver can see the parking space image information and select the parking space he wants on the human–computer interaction interface. The workflow of the entire automatic parking system is shown in Figure 8.

## 4. Vehicle Testing

### 4.1. Simulation Model-Based Experiment

To demonstrate the proposed automatic parking system, a simulation model is built through the Simulink. The simulation of the automatic parking system is conducted under nonparallel initial state conditions based on the actual parameters of the test vehicle. It is shown in Table 2.

The simulation model is shown in Figure 9, which is mainly for trajectory generation evaluation. The simulation results with different initial attitude angle conditions are shown in Figure 10. Figure 10a–d shows the simulation results with different initial attitude angles of the vehicle including $-15^{\circ }$, $-8^{\circ }$, $0^{\circ }$, and $7^{\circ }$. It can be seen that the proposed trajectory generation method can satisfy the parking requirement with nonparallel initial state.

### 4.2. Real Vehicle Experiment

In Equation (5), $L_{ref}$, which is introduced as the referencing target value in the data fusion step, is related to recognition error. It is mainly affected by two factors: driving speed and cross range. To evaluate two factors, all experiments are conducted in a virtual environment as shown in Figure 11, where the ultrasonic sensors are mounted 70 cm above the ground in a 4.6 m×1.8 m vehicle. The role of the stake not only simulates the parking spot, but also represents other vehicles existing, which means if the target vehicle strikes the stake, the experiment is a failure.

We first evaluate driving speed and cross range independently. Also, both the single sensor and the average of double sensors, respectively, are used to do the experiment. Moreover, considering the influence of the other factors, we must ensure that the sensors are stable during the experiment.

First, keeping driving speed at 5 km/h and target parking space length at 6.35 m, seven different cross ranges (0.8 m, 1.0 m, 1.2 m, 1.4 m, 1.6 m, 1.8 m, and 2.0 m) between the target parking space and vehicle are tested. The experiment is repeated 10 times for each condition. We compute the average error of the 10 groups for each distance as the final error. Figure 12a shows the experiment results including single sensor and average of double sensors. It can be seen that they have similar curves.

Then, the cross range is kept at 1.0 m and target parking space length at 6.35 m. Six different driving speed (2 km/h, 3 km/h, 4 km/h, 5 km/h, 6 km/h, and 7 km/h) are tested. The experiment is repeated 10 times for each driving speed. We compute the average error of the 10 groups for each driving speed as the final error. Figure 12b shows the experiment results including single sensor and average of double sensors. It can be seen that recognition error approximately keeps a linear relationship with driving speed for both methods.

Figure 12 shows that both driving speed and cross range have a nearly linear influence on parking spot length error. Based on the analysis and synthesizing the linear influence of the two factors, we formalize a linear formula for error correction.

From the comprehensive factors experiment perspective, we consider the average error of double sensors to be the final measured value. The multilevel design includes seven different cross ranges (0.8 m, 1.0 m, 1.2 m, 1.4 m, 1.6 m, 1.8 m, and 2.0 m) and six different driving speeds (2 km/h, 3 km/h, 4 km/h, 5 km/h, 6 km/h, and 7 km/h). A total of 42 groups of experiments are conducted. Each group of experiment is repeated three times and the average value is calculated. Based on the experiment results, a 3D error curve is obtained as shown in Figure 13. Here, T indicates the cross range. $v_{run}$ and $\delta_{avg}$ are the driving speed and average error of double sensors, respectively.

The value measured by the sensor is usually affected by external conditions, so the sensor requires a separate correction equation. In mathematical models, multiple regression models [40] that can consider surrounding environmental factors have been used to establish relevant correction formulas. According to the results, the mathematical model of multiple linear regression is given as follows:(20)$$y=0.150196x_{1}+0.010583x_{2}-0.2452$$
where y is the measured average error of double sensors; and $x_{1}$ and $x_{2}$ are the horizontal distance and driving speed, respectively. To assure scientific rationality of error correction, $x_{1}$ is the average value of cross range through the upper and lower edges of the target space, and $x_{2}$ is the average driving speed through the target space.

To verify the validity of the regression formula, we use F-value and present the results in Table 3. When α is at level 0.05, the F-value from Equation (20) is 311.3356, and $F_{2,39,0.95}=4.08$. The results show that if F-value is greater than $F_{m,n-m-1,1-\alpha }$, the regression function is significant at level 0.05. Moreover, the residual standard deviation (σ) is 0.016294, and 2σ is 0.032588. Therefore, 95% deviations of the detected errors are within 0.032588 m with this regression function and satisfy the experimental requirements.

## 5. Experimental Results and Discussion

### 5.1. Analysis and Comparison

We next aim to improve the success parking space recognition rate under the simulated parking environment shown in Figure 10. We compare our multi-sensor information ensemble method with single sensor application and the average of double sensors. It should be noted that the cross range is the average value of measured distance through the edges of the target space, and the driving speed is the average value of the speed recorded while driving through the target space.

The experimental results of the single sensor application are shown in Figure 14. The x-axis and y-axis represent the number of experiments and measured length of the target space, respectively. It can be seen that nine groups of experiment results are larger than the target data (6.35 m). The success rate of recognition is approximately 45%, and the maximum recognition error is about 15 cm.

In the double sensor method experiments, the ranging sensors are located on the right-hand side of the vehicle. The driver controls the speed and cross range at a relatively stable level. The experimental process is the same as that of single sensor application. Figure 15 shows the measured and reference data obtained by Equation (20).

Figure 16 shows the average of data of the double sensors shown in Figure 15. The x-axis and y-axis represent the number of experiments and measured data, respectively. As shown in Figure 16, 32 groups of experiment results are above the expected target value of 6.35 m. The success rate of the average method is 64%, and the recognition error is within 9 cm, which means recognition results are better than those obtained with the single sensor method.

Our proposed multi-sensor information ensemble method results are shown in Figure 17. The x-axis and y-axis represent the number of experiments and measured data, respectively. As shown in Figure 16, 47 groups of experimental results are greater than the expected value of 6.35 m. The success rate of multi-sensor information ensemble method is 94%, and the recognition error is within 5 cm, which means our proposed method is better than the single sensor and average of double sensors methods. The experimental results are summarized in Table 4.

The comparison results shows that our proposed multi-sensor information ensemble method can enhance the success rate and reduce recognition error in parking space recognition.

### 5.2. Parking Space Matching and Final Auto-Parking Test Results

For the automatic parking system testing, the parking space recognition, parking space matching, and trajectory generation algorithm are combined. The whole process is shown in Figure 18.

While parking the vehicle, the image sensor recognizes the parking space. The virtual space is then established by parking space matching algorithm. A number of parking space matching tests are performed with different parking initial vehicle states. The parking space matching result is shown in Figure 19.

The longitudinal and horizontal distance between the vehicle and the parking space can affect the trajectory generation. If the size of parking spaces is fixed, the length and width of the parking space can be determined. Therefore, only the coordinates of the red point (reference point) in Figure 18 should be confirmed. The longitudinal and horizontal distance between the vehicle and the parking space can then be calculated based on the confirmed reference point. We measure the reference point 10 times and compare errors between the actual coordinates and measured results as shown in Table 5. It can be viewed that the errors on x and y directions are within 4 cm and 5 cm, respectively, which satisfy the parking requirement.

The trajectory generation algorithm can directly reflect the success parking rate. After finishing parking, we can determine the effectiveness of the generated trajectory from the attitude angle of the vehicle. During the parking process, if no collisions occur and the attitude angle of the vehicle is within $\pm 5^{\circ }$, the parking can be considered to be successful. From the numerous parking experiments, 10 experimental results are randomly selected and shown in Table 5. The left, right, front, and rear columns show the respective distances from the parking space when the vehicle is parked in the parking space.

In Table 6, the initial attitude angle of the vehicle is within $15^{\circ }$. The parking process is a success when the angle is within $9^{\circ }$. Whereas it may fail when the initial angle is $15^{\circ }$. The experimental results show that the proposed automatic parking system algorithms are successful and effectively solve the collision problem during the parking process. The first nine experiments in which attitude angle is within $\pm 15^{\circ }$ are successful with no collisions when the vehicle is parked in the parking space. Only one experiment is unsuccessful because the attitude angle is about $15^{\circ }$. Therefore, the success rate of parking is 90% with the proposed methods.

## 6. Conclusions

This paper has developed the parking space recognition, parking space matching, and trajectory generation-based approach for automatic parking system, which successfully overcomes the drawbacks associated with existing traditional methods. The proposed approach significantly improves the parking performance. In particular, we propose multi-sensor information ensemble algorithm for parking space recognition. Then, the linear mapping is applied to match the parking space. Subsequently, the nonparallel initial state-based trajectory generation algorithm is investigated. Simulation and real vehicle experimental results have been conducted to demonstrate the superior performance with respect to the accuracy parking performance. In detail, the success rate of parking space recognition reaches 94% and its identification error is within ±5 cm. When the initial angle of the vehicle is within $\pm 15^{\circ }$ and the length and width of the parking space are 6 m and 2.4 m, respectively, the parking success rate can reach 90%.

## Figures and Tables

**Figure 1 sensors-21-02261-f001:**
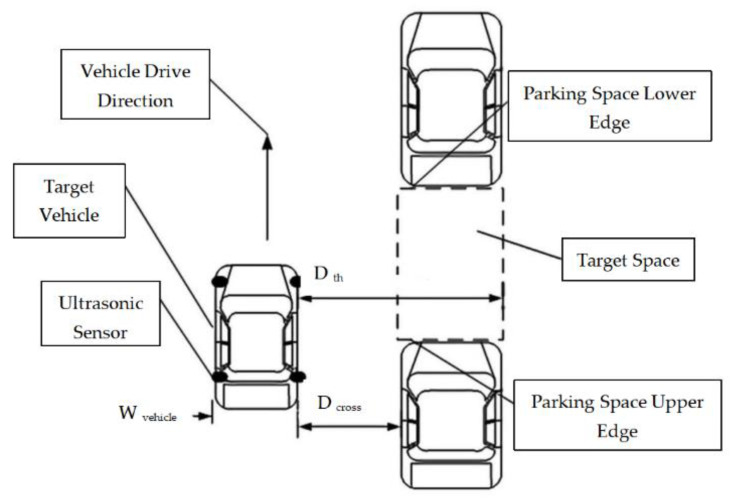
Mimic diagram of parallel parking spot.

**Figure 2 sensors-21-02261-f002:**
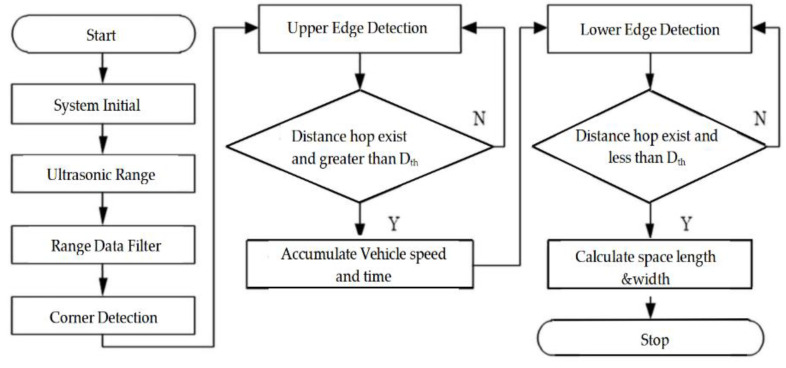
The structure of parking spot detection algorithm.

**Figure 3 sensors-21-02261-f003:**
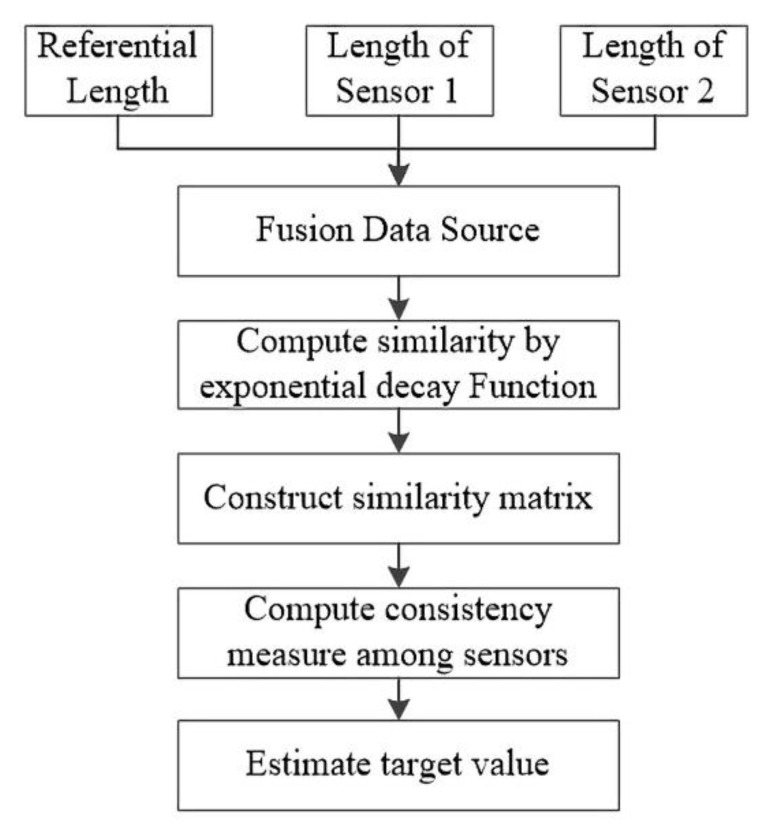
Flowchart on multi-sensor information ensemble.

**Figure 4 sensors-21-02261-f004:**
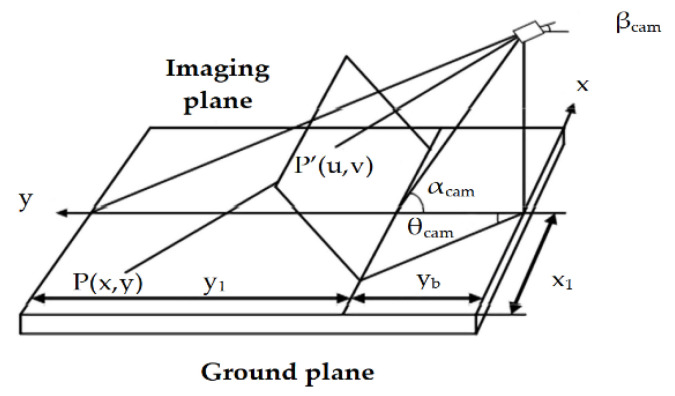
Mimic diagram of parking spot.

**Figure 5 sensors-21-02261-f005:**
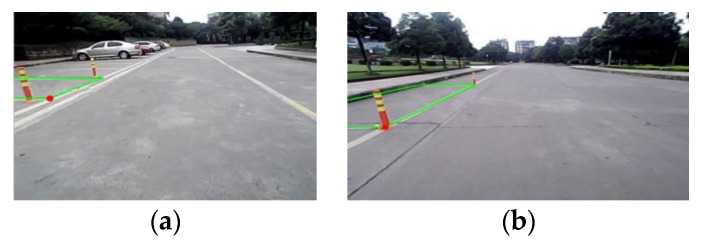
Matching of vertical and parallel parking spaces: (**a**) Matching on vertical parking space, (**b**) Matching on parallel parking space.

**Figure 6 sensors-21-02261-f006:**
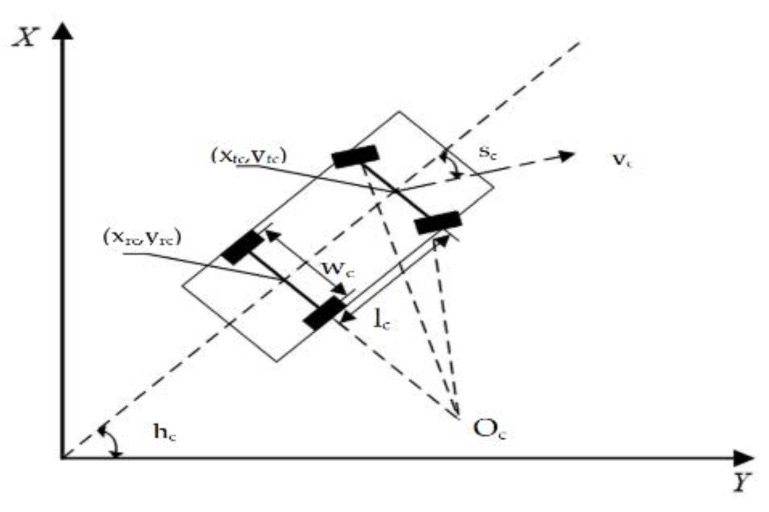
Kinematics model of vehicle.

**Figure 7 sensors-21-02261-f007:**
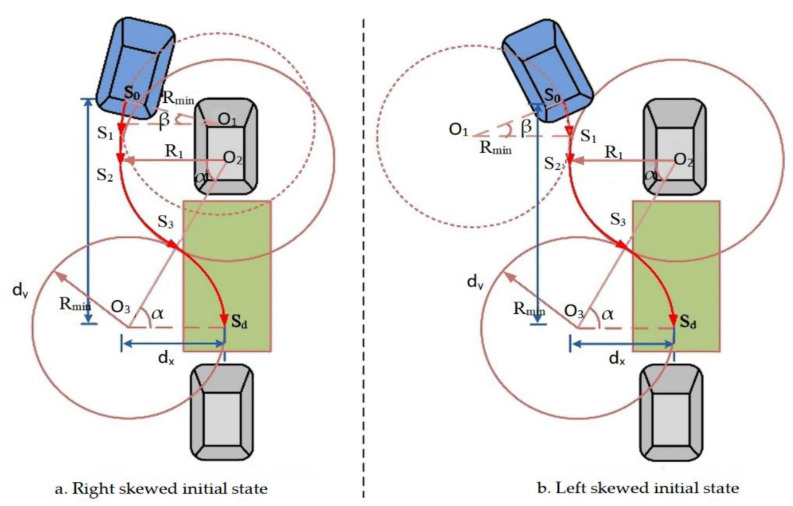
Schematic diagram of nonparallel initial state parking reference trajectory.

**Figure 8 sensors-21-02261-f008:**
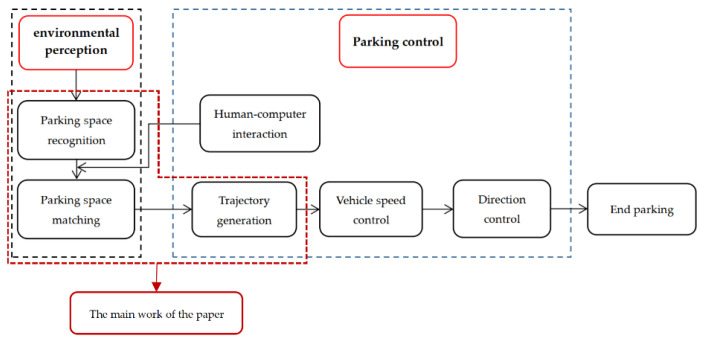
The workflow of the automatic parking system.

**Figure 9 sensors-21-02261-f009:**
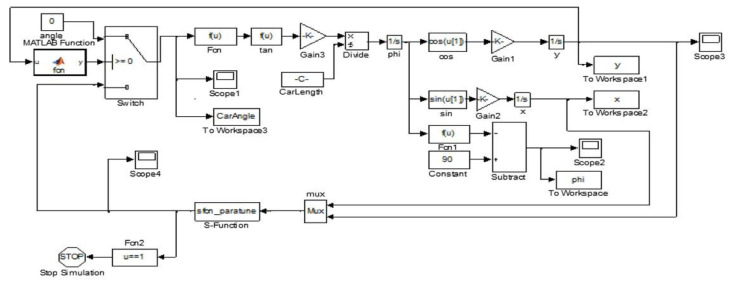
Simulation model of the automatic parking system in Simulink.

**Figure 10 sensors-21-02261-f010:**
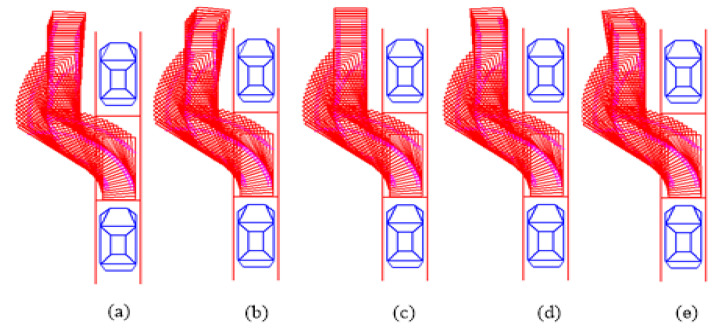
Parking simulation results under nonparallel initial state condition. (**a**–**e**) respectively are the simulation results of different initial conditions

**Figure 11 sensors-21-02261-f011:**
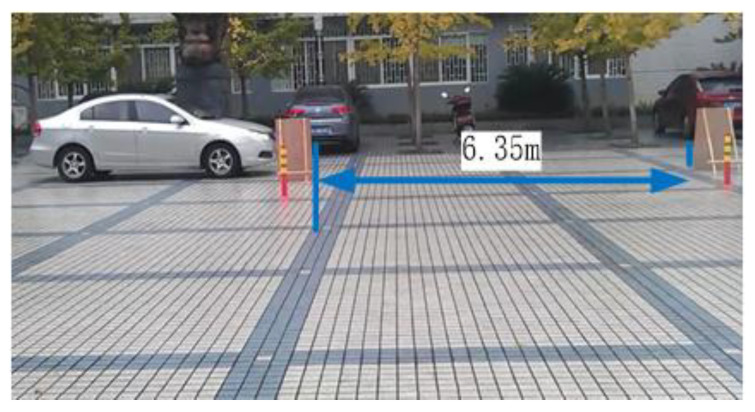
Virtual parallel parking environment.

**Figure 12 sensors-21-02261-f012:**
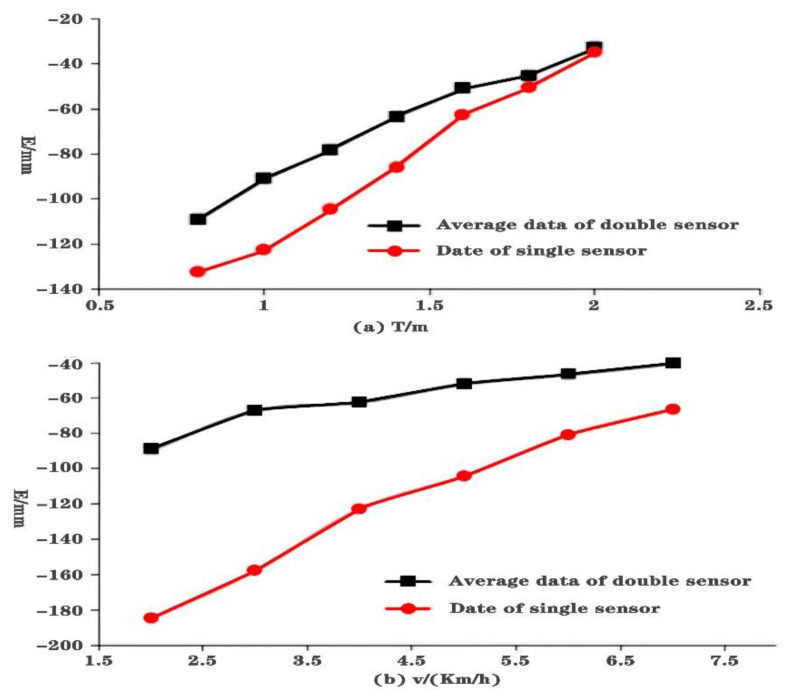
Results on single factor experiment: (**a**) Results on fixed driving speed, (**b**) Results on fixed cross range.

**Figure 13 sensors-21-02261-f013:**
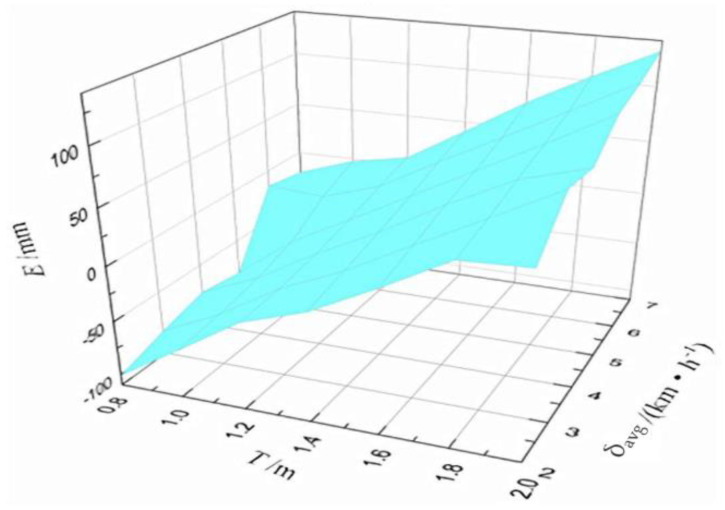
Error curved surface of orthogonal data.

**Figure 14 sensors-21-02261-f014:**
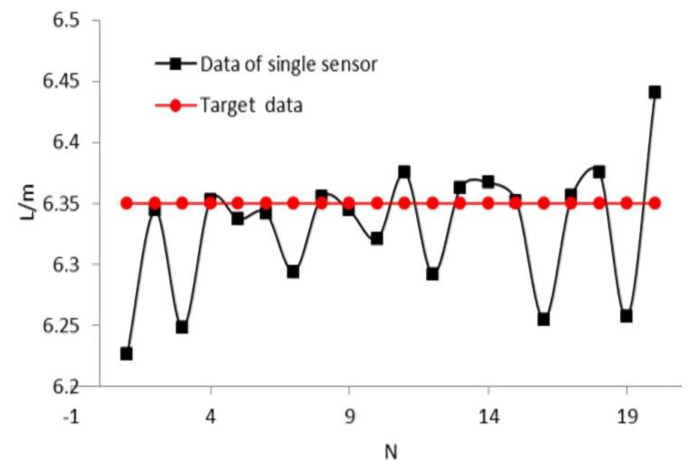
Experimental results of single sensor application.

**Figure 15 sensors-21-02261-f015:**
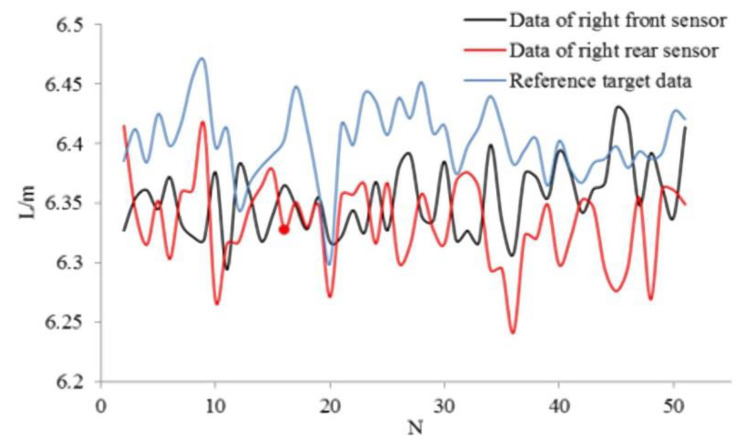
Recognition data of two sensors and reference target data.

**Figure 16 sensors-21-02261-f016:**
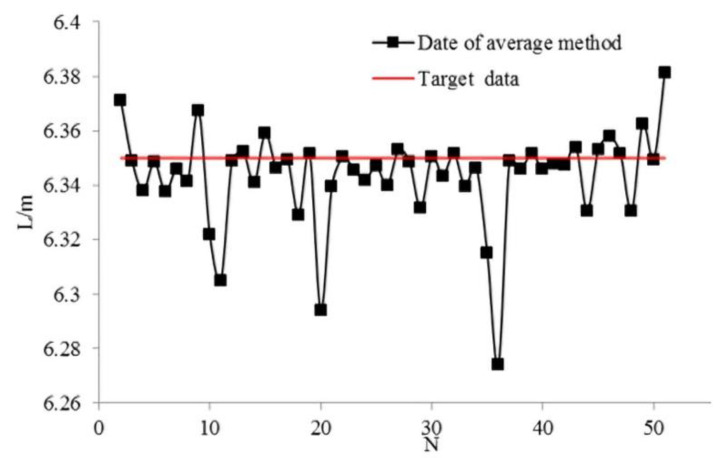
Results of average of double sensors method.

**Figure 17 sensors-21-02261-f017:**
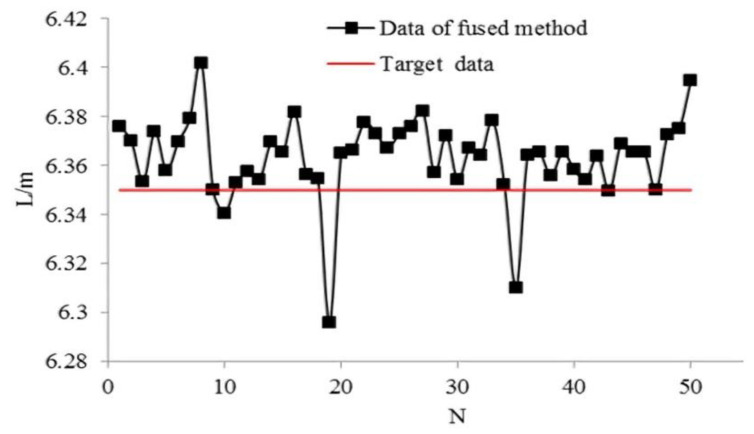
Results of multi-sensor information ensemble.

**Figure 18 sensors-21-02261-f018:**
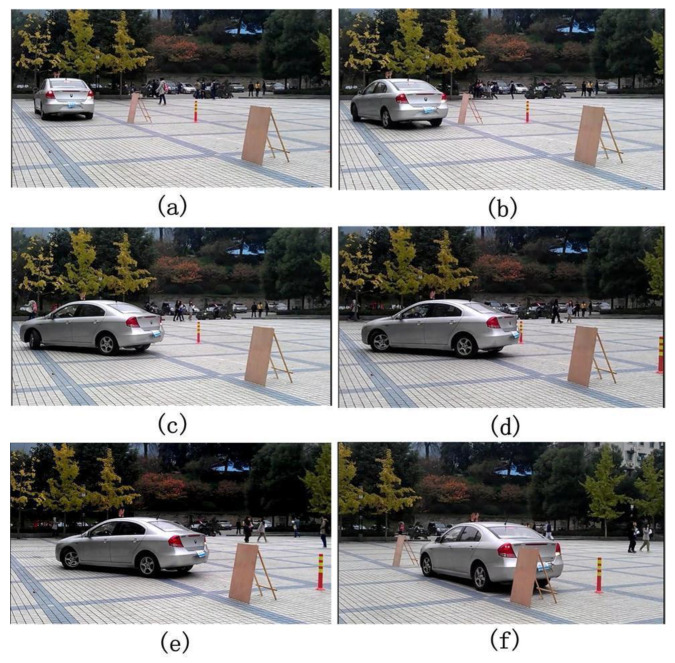
Parking process using proposed automatic parking system. (**a**–**f**) are the actual vehicle test results with different initial conditions

**Figure 19 sensors-21-02261-f019:**
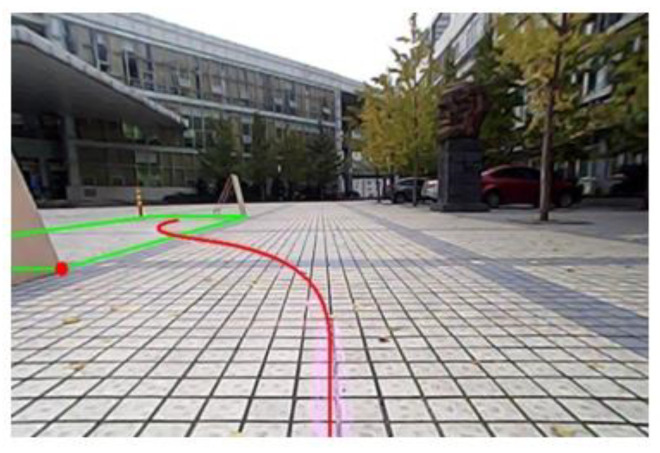
Parking space matching and trajectory generation.

**Table 1 sensors-21-02261-t001:** Performance comparison with other state-of-the-art methods on the various datasets.

Number	Pixel (U,V)	X [cm]	Y [cm]	Calculated Distance [cm]	Actual Distance [cm]	Relative Error
1	(168,478)	24.3	54	59.2	56.5	4.8
2	(228,305)	37.9	151	155.7	150.3	3.6
3	(140,258)	118.8	245	272	268.2	1.4
4	(83,235)	218.2	344	407.4	402.7	1.2
5	(245,222)	87.7	443	451.6	446.4	1.3
6	(196,212)	184.9	564	593.5	586.8	1.1
7	(88,206)	413.7	673	789.7	781.2	1.1
8	(76,203)	481.3	745	885.9	877.1	1.2
9	(273,201)	97.88	801	806.9	796.6	1.3
10	(115,198)	489.4	904	1027.9	1012	1.5

**Table 2 sensors-21-02261-t002:** Simulation parameters [mm].

Vehicle length	4600
Vehicle width	1800
Front-wheel gauge	1500
Rear-wheel gauge	1487
Wheelbase Front overhang	2650
Front overhang	970
Rear overhang	980
Minimum turning radius	4200
Parking space length	600
Parking space width	240

**Table 3 sensors-21-02261-t003:** Test of multiple linear regression model.

α	F-Value	F_0.95_ (2,39)	Results
0.05	311.3356	4.08	significant

**Table 4 sensors-21-02261-t004:** Comparison results on parking space recognition.

Method	Times	Success Rate	Max. Recognition Error [cm]
Single sensor	20	45%	15
Double sensors	50	64%	9
Ours	50	94%	5

**Table 5 sensors-21-02261-t005:** Range data of reference points.

Num	Reference Point	Actual Coordinates [cm]	Measured Coordinates [cm]
1	P0	(89, 187)	(86, 190)
2	P0	(76, 195)	(80, 198)
3	P0	(103, 229)	(100, 230)
4	P0	(95, 234)	(94, 239)
5	P0	(62, 255)	(66, 258)
6	P0	(123, 278)	(119, 282)
7	P0	(115, 194)	(114, 192)
8	P0	(134, 298)	(137, 301)
9	P0	(89, 238)	(92, 241)
10	P0	(154, 219)	(151, 214)

**Table 6 sensors-21-02261-t006:** Parking results under nonparallel initial state and unequal radii conditions.

Number	Collision	Left [cm]	Right [cm]	Front [cm]	Rear [cm]	Angle [degree]	Results
1	NO	23	26	82	56	1	Success
2	NO	24	25	86	55	3	Success
3	NO	22	27	85	57	4	Success
4	NO	29	20	88	52	−4	Success
5	NO	25	26	84	56	5	Success
6	NO	28	23	86	55	−5	Success
7	NO	29	18	88	52	−7	Success
8	NO	23	26	86	55	9	Success
9	NO	24	25	81	59	15	Success
10	NO	38	12	88	52	−15	Failure

## Data Availability

Not applicable.

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
