# Peer review of "Multi-Sensor Information Ensemble-Based Automatic Parking System for Vehicle Parallel/Nonparallel Initial State"

_sensors, 2021, doi:10.3390/s21072261_

Round 1

Reviewer 1 Report

This paper proposes an automatic parking system for vehicle parallel/nonparallel initial state. It uses multiple sensors' information to improve the performance. However, there are some problems to be clarified.

  1. Which the sensors exactly are used? Camera and Ultrasonic? what is the type, and other parameters? It is important for understanding.
  2. The paper is not well written enough. The diagram and other figures are not clear(with too low quality). Some abbreviation, such as APS, is not given the full name.
  3. The experiments are difficult to reproduce. Please detail them clearly.
  4. Why 6.35m is selected? it is not a standard slot.
  5. some reference about parking detection should be survey or compared. Such as,
    1. Vision-Based Parking-Slot Detection: A DCNN-Based Approach and a Large-Scale Benchmark Dataset

    2. Geometric Features-Based Parking Slot Detection.

    3. Parking Slot Detection on Around-View Images Using DCNN

Round 2

Reviewer 1 Report

Most of my comments are responded well. I just have some minor concerns.

  1. The diagram of fig 8 is not very good. It should emphasize your work inside. Currently, it looks all the blocks are your contribution.
  2.  In my last comment 5, I search for some related reference and I suggest putting them into your related work or in comparison. I can understand it may be not suitable for comparison or into the related work. However, why do you put so many other reference papers into the new version?  It is difficult to judge your contribution, if new reference papers are included. As you may understand, I should judge if my previous comments are answered well and I will not put new comments in the second round reviewer. However, new references are included, I cannot judge. Hence, please list the reason for each new included paper.

Reviewer 2 Report

See the attached document

Round 3

Reviewer 1 Report

I have no more concerns.

Author Response

Thank you very much for your review
